# Dynamic Gait Analysis in Paediatric Flatfeet: Unveiling Biomechanical Insights for Diagnosis and Treatment

**DOI:** 10.3390/children11050604

**Published:** 2024-05-17

**Authors:** Harald Böhm, Julie Stebbins, Alpesh Kothari, Chakravarthy Ughandar Dussa

**Affiliations:** 1Orthopaedic Hospital for Children, Treatment Center Aschau im Chiemgau, 83229 Aschau im Chiemgau, Germany; 2Faculty of Engineering and Health Göttingen, University of Applied Sciences and Arts, 37077 Göttingen, Germany; 3Oxford University Hospitals NHSFT, Oxford OX3 9DU, UK; julie.stebbins@ndorms.ox.ac.uk (J.S.); alpesh.kothari@ndorms.ox.ac.uk (A.K.); 4Department of Orthopaedic Trauma and Surgery, Friedrich-Alexander University Erlangen, 91054 Erlangen, Germany; dussacu1@web.de

**Keywords:** kinematics, kinetics, multi segment foot model, walking, pes planus, flexible flatfoot

## Abstract

Background: Flatfeet in children are common, causing concern for parents due to potential symptoms. Technological advances, like 3D foot kinematic analysis, have revolutionized assessment. This review examined 3D assessments in paediatric idiopathic flexible flat feet (FFF). Methods: Searches focused on paediatric idiopathic FFF in PubMed, Web of Science, and SCOPUS. Inclusion criteria required 3D kinematic and/or kinetic analysis during posture or locomotion, excluding non-idiopathic cases, adult feet, and studies solely on pedobarography or radiographs. Results: Twenty-four studies met the criteria. Kinematic and kinetic differences between FFF and typical feet during gait were outlined, with frontal plane deviations like hindfoot eversion and forefoot supination, alongside decreased second peak vertical GRF. Dynamic foot classification surpassed static assessments, revealing varied movement patterns within FFF. Associations between gait characteristics and clinical measures like pain symptoms and quality of life were explored. Interventions varied, with orthoses reducing ankle eversion and knee and hip abductor moments during gait, while arthroereisis normalized calcaneal alignment and hindfoot eversion. Conclusions: This review synthesises research on 3D kinematics and kinetics in paediatric idiopathic FFF, offering insights for intervention strategies and further research.

## 1. Introduction

Flexible flatfoot (FFF) is a common foot condition among paediatric populations and is often the cause of visits to specialized clinics [1]. This deformity is characterized by subtalar valgus, longitudinal sag at the talonavicular joint, and midtarsal abduction. However, the different components of this deformity may vary in severity in individuals and thereby result in several expressions of the same deformity. Although a flat arch is typically observed at birth, it often develops into a normal concave arch by the age of 5–6 years [2,3]. However, approximately 24% of school-aged children continue to exhibit persistent flatfoot, potentially necessitating therapeutic interventions [3]. Between 10% and 60% of children with PFF experience symptoms such as pain, fatigue, and reduced sports performance [4,5]. For those whose symptoms do not respond to conservative treatment, surgery may be necessary [1,6].

While flatfeet can have a multifactorial aetiology, including neurologic disorders [7], overcorrected clubfeet [8], and syndromes [9], narrowing the focus to idiopathic aetiology enables researchers to delve into the unique biomechanical aberrations inherent to this major and most common subgroup. This targeted approach fosters a more comprehensive understanding of the condition.

For the assessment and planning of treatment of foot deformities, standardized standing X-rays are typically used, supplemented by 3D CT scans when necessary [10,11]. However, since FFF is primarily a dynamic issue, static imaging is unable to fully capture the underlying clinical problem [1]. Dynamic pedobarography, a common tool in diagnosing paediatric flat feet, offers dynamic quantitative data on pressure distribution during movement [12]. The commonly utilized arch index is computed by comparing the pressure area of the midfoot to the combined areas of the forefoot, midfoot, and hindfoot. While useful, this 2D measurement overlooks the intricate nature of three-dimensional foot deformities and their impact on movement functionality.

Dynamic 3D motion analysis is a non-invasive modality for assessing foot dynamics during various locomotor tasks, providing invaluable insights into the complexities of foot function in paediatric patients with flatfeet. By employing state-of-the-art motion capture technologies, researchers can meticulously examine the biomechanical nuances associated with flatfoot deformities, shedding light on subtle alterations in gait patterns, Ground Reaction Forces (GRF), joint moments, and foot kinematics.

Three-dimensional marker-based foot models complement foot diagnostics, measuring complex movements between foot segments during walking and identifying pathological movements or patterns often overlooked by static assessments [13]. Meeting key criteria of reliability, validity, and responsiveness to clinical interventions, these models have been continuously developed and optimized for paediatric FFF applications, demonstrating sensitivity to surgical interventions [5,14].

In this systematic review, we aim to explore studies reporting 3D kinematic and kinetic data in flatfeet, particularly multi-segment foot kinematics, as valuable tools for evaluating paediatric flatfeet. Through a synthesis of the current literature and clinical insights, we will examine how gait analysis has been used to elucidate biomechanical complexities underlying flatfoot deformities, implications for diagnosis and treatment planning, and potential avenues for future research.

## 2. Methods

### 2.1. Inclusion and Exclusion Criteria

This review focused on studies of children or adolescents diagnosed with idiopathic FFF. Studies assessing adult feet, non-idiopathic cases like FFF due to syndromes or neurological diseases, bony coalitions, lower limb surgery patients, and those with FFF due to excessive body weight were excluded. Included studies were required to offer a 3D analysis of kinematics and/or kinetics during posture or locomotion, excluding those solely analysing pedobarography or radiographs. The review considered only recent articles published between 2010 and 2023.

### 2.2. Search Strategy

The search was carried out in the following databases: PubMed, Web of Science, and SCOPUS. The following medical subject headings (MeSH) were used: flatfoot, paediatrics, and child, according to the characteristics of each database, accompanied by the Boolean operators “AND” and “OR”. The following search strategy was used: ((“Flatfoot”[Mesh] AND (“Paediatrics”[Mesh] OR “Child”[Mesh] OR “Child, Preschool”[Mesh]) AND “Gait Analysis”[Mesh]) OR ((“Flexible Flatfoot”[tw] OR “Flat Foot”[tw] OR “Pes Planus”[tw] OR Flatfoot[tw] OR “Foot, Flat”[tw] OR “Flatfoot, Flexible”[tw]) AND (Paediatrics[tw] OR “Preschool Child*”[tw] OR Child*[tw] OR “Child*, Preschool”[tw]) AND (“Gait Analysis”[tw] OR “3D-Analysis”[tw] OR “Foot Model”[tw] OR “Foot Kinematics”[tw] OR “Kinetics”]))).

In addition, the papers’ bibliographies were reviewed. The search term [tw] stands for “textword”, which refers to the title, abstract, and author-provided keywords.

### 2.3. Study Selection

Two authors, HB and UD, conducted the initial study selection process. Duplicates were removed following database searches. Titles and abstracts were screened based on predetermined criteria. Selected studies were thoroughly reviewed to ensure eligibility. JS and AK then assessed the remaining papers for compliance and identified any relevant gaps in the literature.

### 2.4. Data Extraction and Management

To meet the study objectives, data extraction focused on key elements from selected studies, including age and recruitment details of the FFF sample. Additionally, information on instrumentation, biomechanical models, movement, and footwear was extracted. The data extraction process also involved assessing study goals and identifying the main findings related to kinematics and kinetics. Quality assurance procedures were rigorously implemented throughout the review process to uphold the accuracy and reliability of the provided summaries. The extracted data underwent synthesis into comprehensive summaries using narrative synthesis, led by one author, HB. Subsequently, the remaining three authors independently reviewed and cross-checked the summaries.

## 3. Results

Using the outlined search strategy, we initially identified 134 studies from the databases, along with 3 additional records sourced from reference lists of retrieved papers. After removing 64 duplicated records, 70 studies remained, subject to evaluation based on title and abstract by two independent reviewers. Out of these, 44 were excluded due to discrepancies in inclusion criteria, such as lacking kinematic or kinetic data or involving participants outside the scope of children. Subsequently, 26 full texts underwent eligibility assessment, resulting in the exclusion of two studies: one due to insufficient explanation of calculated kinematic parameters [15] and the other for presenting incomplete kinematic waveforms without statistical or descriptive evaluation [16]. Thus, a total of 24 papers fully met the inclusion criteria. Refer to Figure 1 for the PRISMA flow diagram illustrating the study selection process and to Table 1 for details on the 24 included studies.

### 3.1. Population

Children were recruited from various sources, including paediatric orthopaedic clinics, elementary schools, and the community. Those recruited from orthopaedic clinics typically exhibited symptoms, which may have been the primary reason for seeking medical attention [1]. This set them apart from those recruited from schools or the community, who were predominantly identified by their low arch. In these cases, participants’ feet were assessed for a navicular drop greater than 10 mm [41], greater than 4° of eversion in calcaneal stance position [42], or an arch height index less than 0.31 [43].

Most studies involved children over the age of 5, as it has been well-established that by this age, most children have developed an arch, the fat pad has been absorbed, and all foot bones show ossification [3]. An exception to this trend was observed in the study conducted by Krautwurst et al., which included children as young as 3 years old [28]. The mean age across the majority of studies was around 11 years, with a standard deviation ranging between 1 and 2 years.

### 3.2. Technology and Human Model

Marker-based assessment was predominantly utilized across the studies, except for one employing electromagnetic tracking [19]. Regarding foot kinematics, two multi-segment models were employed. The Oxford Foot Model (OFM), consisting of hindfoot, forefoot, and hallux segments, was utilized in 11 studies, whilst the Rizzoli model, incorporating an additional segment (calcaneus, midfoot, forefoot, and hallux), was utilized in three studies [14,23,31]. Additionally, the Heidelberg method for calculating foot angles was applied in two studies [17,28]. The study employing electromagnetic tracking reported hindfoot angles during walking without further details on how they were calculated [19].

Lower body angles and kinetics were predominantly calculated using the Plug-in Gait model (PiG), with nine studies employing this method. Among them, four studies combined PiG with the OFM. Additionally, two studies utilized the Helen Hayes Method implemented in Orthotrak software without specifying a reference or providing further information. The study employing electromagnetic tracking did not offer detailed model information. An illustrative example of a patient with FFF utilizing markers to assess the PiG alongside the OFM is depicted in Figure 2.

### 3.3. Footwear and Movement

Barefoot assessment was conducted in the majority of studies, with only three opting to use standardized shoes for all participants [19,29,32]. In most cases, participants were instructed to walk at their self-selected speed. One study focused on analysing the 3D motion of the hindfoot during the heel raise test [28], while another measured static foot deformity under weight-bearing conditions [24].

### 3.4. Purpose and Main Outcome

The studies’ objectives encompass four key areas of investigation.

Comparative analysis between FFF and Typically Developed Feet (TDF);Classification of foot types;Examination of the relationship between foot characteristics and clinical measures;Assessment of the effectiveness of interventions.

The subsequent section provides a comprehensive summary of the primary outcomes within these categories.

#### 3.4.1. Comparative Analysis between Flexible Flat Feet and Typically Developed Feet

The investigation into foot kinematics during walking has revealed distinct disparities between FFF and TDF across all three planes of motion, as well as in the hindfoot-to-tibia and forefoot-to-hindfoot relationships. In the sagittal plane, stance phase analysis during walking revealed diminished peak dorsiflexion of the hindfoot-to-tibia alongside heightened midfoot dorsiflexion [5,20,27,34,35] and a lower medial arch throughout stance [31]. In the frontal plane, increased peak hindfoot valgus [5,29,31,32,33,34] and relative forefoot supination [5,23,24,29,31,32,33,34] were observed. In addition, in the transverse plane, increased forefoot abduction during walking [5,23,29,31,32,33] and standing [24] were noted as well as a reduced navicular drift [22]. An illustrative depiction of severely affected FFF waveforms is presented in Figure 3.

Shod walking and electromagnetic tracking failed to discern differences in hindfoot angles between FFF and TDF. However, this could be due to limitations posed by electromagnetic tracing, shod walking, or due to mild flatfoot deformity in the recruited subjects.

When comparing the results of a multi-segment foot model to a single-segment model, notable differences emerge in foot kinematics. Specifically, these differences are apparent in flatfeet but not in cases of TDF [27]. The calculations of the single-segment model tend to overstate ankle joint varus and dorsiflexion in flatfeet, contradicting the established diagnosis.

Lower body kinematics in the sagittal plane exhibited a reduced range of plantarflexion during push-off [30,39] and increased knee flexion [30,33]. Notably, frontal plane dynamics indicated heightened knee valgus [26,33], although the presence of knee valgus has been contested [31,37]. Furthermore, in the transverse plane, increased knee rotation ranges and increased peak internal rotation were observed [19]. Additionally, pelvis external rotation in a late stance correlated with a flatter foot posture [26]. In two studies, an increased foot external progression angle was observed before patients underwent calcaneal lengthening procedures. Following surgery, the foot progression angle was normalized [36,40].

Analysis of leg kinetics highlighted a significant association between flatfoot posture and a reduction in the second peak of the vertical GRF [26,30,31,32,36,39], with a concomitant increase in the first peak [31,39]. Moreover, late stance hip extension and knee varus and rotation moments were diminished [26], while mean ankle moment and power during push-off attenuated [30]. In the frontal plane, the knee abduction moment (KAM) was smaller in FFF [37].

#### 3.4.2. Classification of Foot Types

The distinction between populations of FFF and TDF has often relied on measuring navicular drop during standing [17,18,29,32], as detailed in Section 3.1. However, the study by Boryczka-Treffler et al. [39] raised questions about whether this classification should occur during walking rather than static standing. Notably, Boryczka et al. demonstrated that only dynamic classification revealed significant kinematic differences between FFF and TDF [39]. Furthermore, within the FFF group, Böhm et al. conducted classification based on foot kinematics [34]. Utilizing PCA and cluster analysis, they identified two distinct movement patterns. Pattern 1 exhibited characteristics such as a flat foot with deviations across all planes, including hindfoot inversion, lowered longitudinal arch, and forefoot abduction. In contrast, Pattern 2 represented a seemingly normal foot movement, despite a clinically examined and/or X-ray evidence of flat foot pathology during standing. This suggests potential muscular compensation during walking to counteract static deformity. Notably, a key distinguishing feature between the patterns was hindfoot inversion at foot strike.

#### 3.4.3. Relation with Clinical Measures

Gait kinematics and kinetics have been related to various clinical measures, including symptoms (with a focus on pain in particular), radiography, and quality of life. Hösl et al. [20] compared symptomatic and asymptomatic individuals and noted no differences in foot kinematics but observed reductions in push-off energy. Conversely, Kerr et al. [24,33] reported increased forefoot abduction during walking and standing, along with hindfoot eversion during standing in symptomatic feet.

Three studies focussed on pain as a symptom [5,26,28]. While Krautwurst et al. found no relationship between foot pain and heel raise test motion [28] and Kothari et al. found no relation between flatfoot posture and proximal joint symptoms [26], Böhm and Dussa identified a correlation between calcaneal lateral shift and pain [5]. This correlation was further supported by the reduction in pain following surgery. Quality of life scores, with questions pertaining to pain, were strongly associated with increased hindfoot eversion and forefoot supination [25].

Regarding radiographic angles, Böhm et al. [35] found little to no relationship between foot kinematics and corresponding angles on lateral or anterior–posterior radiographs during relaxed standing. In one study, Portinario et al. [23] utilized visual observation of hindfoot valgus to refine marker placement in a foot model, aligning it more closely with clinical observations.

#### 3.4.4. Effects of Interventions

The studied interventions included orthoses [29,32], arthroereisis [5,14], and calcaneal lengthening osteotomy [36,40]. Arthroereisis was effective in normalizing calcaneal lateral shift [5] and achieving physiological hindfoot alignment [14]. Foot orthoses were shown to decrease ankle eversion moments, knee moments, and hip abductor moments during walking [29]. A four-month treatment with foot orthoses improved peak ankle internal rotation angles and maximum knee external and internal rotation angles [32]. Additionally, the first peak of the GRF was significantly reduced [32].

## 4. Discussion

This review identified 24 studies assessing 3D foot kinematics in paediatric idiopathic FFF. Children were recruited from various settings, with those from orthopaedic clinics typically showing symptoms, while those from schools or communities were mainly identified by low arches. Most studies involved children over 5 years old resulting in an average age of 11 years of most of the studies. Marker-based assessments were predominantly used, with the OFM being the most common. Most studies focused on barefoot assessments, analysing foot kinematics and kinetics during walking. The key study objectives included comparative analysis, foot type classification, examination of foot characteristics and clinical measures, and assessment of intervention effectiveness. Notable findings showed distinct kinematic differences between FFF and TDF across multiple planes, with some studies exploring the relationship between foot pain and specific kinematic parameters.

### 4.1. Population

Children recruited from paediatric orthopaedic clinics may exhibit more severe involvement compared to those from elementary schools as they seek medical attention due to symptoms or significant deviations. Most studies focused on children over the age of 5, a stage at which arch development and the absorption of the fat pad, along with the appearance of ossification centres in the tarsal bone are typically completed [3]. Additionally, conducting gait analysis in small children presents challenges, as placing markers on their tiny feet can be difficult, compounded by the need for patience with young patients.

### 4.2. Technology and Human Model

Marker-based assessment primarily utilized the OFM and PiG models in tandem, which facilitated the comparison of foot and leg kinematics across studies. Several researchers have implemented supplementary optimizations to enhance the reliability and validity of knee joint axis determination [18,38]. The Helen Hayes marker set and model were also utilized to compute leg angles. However, these were all variations in the conventional gait model [21,44]; therefore, it was reasonable to compare data between studies.

While the OFM and the Heidelberg model offered valuable insights into hindfoot-to-forefoot movement, the incorporation of an additional midfoot segment in the Rizzoli model lacks a direct linkage between the forefoot and hindfoot, thereby complicating comparative analyses. Moreover, establishing a direct relationship between the forefoot and hindfoot could be more intuitive for healthcare practitioners, as the foot (excluding talus) is commonly conceptualized as an osseofibrous plate twisting between the hindfoot and forefoot [45].

### 4.3. Footwear and Movement

Barefoot walking at a self-selected pace emerged as the predominant choice, particularly in studies examining detailed multi-segment foot kinematics. The preference for barefoot walking is likely driven by the lack of clearly defined anatomical landmarks on footwear. In addition, relative movement between the foot and shoe may occur. Consequently, such movement may not fully reflect the foot deformity and movements characteristic of FFF. This insight may explain the findings of Shih et al.’s study, which revealed no discernible difference between FFF and TDF during shod walking [19].

### 4.4. Purpose and Main Outcome

#### 4.4.1. Comparative Analysis between Flexible Flat Feet and Typically Developed Feet

The investigation into foot kinematics during walking has shown distinct disparities between FFF and TDF across all three planes of motion in the hindfoot-to-tibia and forefoot-to-hindfoot relationships [20,34]. There is common agreement among studies that hindfoot eversion and forefoot supination are increased in FFF [18,20,25,26,34]. This combination of increased forefoot supination with increased eversion with about the same magnitude can be interpreted as untwisting of the footplate in flatfeet [46]. In addition, the untwisting of the footplate is typically associated with longitudinal arch flattening, tibia internal rotation, and forefoot abduction [46]. However, variations in the extent of eversion and supination between FFF and TDF were noted across various studies. This discrepancy could stem from several factors. Firstly, differences in the demographics of participants, such as whether they were recruited from hospitals or schools, may have contributed to the observed variations. Secondly, individuals who exhibit a flat arch while standing can display differing degrees of flatfoot manifestation during walking, highlighting the complexity and variability of this condition. This discrepancy may be attributed to the distinct biomechanical demands between standing and walking. While the shape of the arch during standing is primarily dictated by bone structure, along with the strength and flexibility of ligaments [47], walking engages foot muscles to stabilize the foot [48]. In cases of flatfeet, altered structural characteristics and properties of ligaments may fail to adequately support the foot during relaxed standing. However, during walking, muscular activation may compensate for these structural deficiencies [34,49]. This notion is supported by findings indicating that dynamic walking characteristics are not always correlated with static standing radiographs for flatfoot deformity [35].

The dynamics of decompensated feet diverge from TDF, primarily manifesting in frontal plane deviations such as hindfoot eversion and forefoot supination. In addition, deviations are observed in the sagittal plane, including midfoot dorsiflexion and hindfoot plantarflexion, as well as increased hindfoot external rotation and forefoot abduction in the transverse plane [34]. These findings strongly support the concept of footplate untwisting, characterized by arch flattening, hindfoot external rotation, and forefoot abduction [46]. Notably, while frontal plane deviations are pronounced, deviations in the sagittal and transverse planes partially overlap with the standard deviations of TDF. Therefore, significant differences were observed only in decompensated feet, highlighting their distinct biomechanical characteristics [34].

Distinguishing between decompensated and compensated feet is paramount for assessing the impact of FFF on knee and hip kinematics. When FFF is compensated by the musculature, exhibiting normal kinematics during walking, deviations at the knee and hip joints attributed to the flatfoot deformity while standing may not necessarily manifest during ambulation. This notion finds support in a study that revealed significant differences in gait parameters between FFF and TDF only when feet are categorized as flat during walking, as opposed to their classification during standing [39]. This discrepancy in defining flat feet may contribute to the challenges in obtaining consistent results regarding leg kinematics. For instance, while the untwisting of the footplate typically involves an expected increase in internal tibial rotation in FFF [46,48], conflicting findings exist during walking. Shish et al. reported increased internal knee rotation [19], whereas Kothari et al. did not observe such alterations [26].

A notable inconsistency arises regarding knee valgus in the frontal plane, as evidenced by findings in [18,26,33], in contrast to the absence of such observations in [31,37]. This discrepancy could be attributed to methodological disparities, including variations in exclusion criteria for knee valgus deformities [37] or demographic factors such as overweight status. Overweight children often exhibit a higher prevalence of knee valgus in conjunction with FFF [50]. Additionally, overweight status may correlate with increased hip external rotation [51], as demonstrated in a study where flat-footed individuals tended to be heavier compared to their TDF counterparts of similar age [17]. Moreover, knee valgus in overweight individuals may potentially be associated with hindfoot eversion and subsequent lateral deviation of the lower limb mechanical axis [37,52].

Analysis of leg kinetics in individuals with flatfoot posture reveals a notable alteration in GRF compared to TDF. Specifically, there is a decrease in the second peak of vertical GRF coupled with an elevation in the first peak. This indicates a compromised capacity of flat feet to function as a rigid lever during propulsion. Such a compromise may arise from reduced hindfoot inversion ability during push-off, resulting in a lack of midfoot rigidity necessary for generating the second peak of GRF [53]. Furthermore, in the frontal plane, there is a smaller knee abduction moment (KAM) observed in flatfoot individuals [37], which may potentially influence growth modulation, as flatfoot and knee valgus often coexist.

#### 4.4.2. Classification of Foot Types

The dynamic mathematical classification differentiates between decompensated and compensated feet, with decompensated feet displaying the most pronounced deviations from TDF [34]. This contrasts with standing radiographic classification, which identifies four deformation types based on hindfoot and forefoot involvement [10]. Particularly noteworthy is the emergence of hindfoot inversion at push-off as a key distinguishing feature among these patterns [34], with the primary involvement of the tibialis posterior and tibialis anterior muscle groups [54]. This compensation during push-off potentially correlates with hindfoot inversion assessed through clinical manoeuvres such as standing on tiptoes [55] and engaging these muscles. This correlation could provide a simple assessment method without requiring a full 3D foot analysis. However, further research is necessary to validate this correlation and replication of the identified clusters by Böhm et al. [34] by other research centres is warranted. The implications of dynamic classification in understanding gait pathologies associated with FFF have been extensively discussed in the preceding Section 4.4.1.

#### 4.4.3. Relation with Clinical Measures

There is a consensus that surgical intervention is warranted for symptomatic FFF cases only after exhausting conservative treatments [55]. Given the pivotal role of symptoms in guiding treatment decisions, numerous studies have endeavoured to differentiate between symptomatic and asymptomatic patients. Among children with FFF, common symptoms include pain, fatigue, and impaired sports performance [56]. Pain may be linked to lateral calcaneal shift during walking [5], with symptoms correlating with abnormal hindfoot eversion and forefoot abduction during standing, as well as forefoot abduction during walking [24,33]. However, Hösl et al. found no discernible differences in foot kinematics [20], possibly due to compensatory muscular adaptations leading to near-normal foot kinematics during walking. However, anatomically restoring foot shape and kinematics in compensated flat feet necessitates heightened muscular activity, thereby increasing susceptibility to dysfunction and injury due to overuse.

In contrast to the relationship with pain, quality of life scores are significantly associated with increased hindfoot eversion and forefoot supination [25]. Quality of life assessments encompass emotional and footwear-related domains in addition to pain and may therefore be different to the assessment of pain alone.

#### 4.4.4. Effects of Interventions

The interventions examined in this study encompassed foot orthoses, arthroereisis, and calcaneal lengthening osteotomy. Each intervention showed distinct effects on foot and lower limb biomechanics, offering valuable insights into their therapeutic potential for addressing various foot-related pathologies. Consequently, the models employed in these studies are sensitive to reveal the effects of conservative and surgical interventions. Since FFF is a 3-dimensional deformity, all planes of motion in the forefoot and hindfoot should be reported to estimate the effect of the specific intervention in a certain plane. Regrettably, forefoot abduction was not reported in the study that reported on the calcaneal lengthening osteotomy [40], despite it being the parameter anticipated to undergo the most significant change.

### 4.5. Challenges and Future Directions

A flexible flat foot presents as a complex three-dimensional deformity with varying severities across different planes [1,10]. Utilizing data derived from 3D kinematic analysis can inform tailored treatment plans, spanning from orthotic prescriptions to physical therapy and surgical interventions. By comprehending individual variations in foot motion, interventions can be precisely targeted, thereby optimizing outcomes for paediatric and adolescent patients with FFF. While the studies reviewed in this analysis primarily identified parameters aiding in the diagnosis of flatfeet and elucidated sources of pain and impacts on quality of life, there is a need for more studies that focus on recommendations regarding therapy in future research endeavours. The acquired data not only aid in diagnostic indications but also facilitate precise surgical planning. For instance, measuring the lateral displacement of the calcaneus from 3D gait analysis directly informs potential surgical interventions, such as a calcaneal medial sliding osteotomy, to address this deformity. Additionally, kinematic deviations may suggest the potential success of minimally invasive arthroereisis in improving sports performance and quality of life. In the context of a painful FFF, with pain arising at the medial arch, a lateral displacement of the calcaneus should be specifically looked for in order to plan a targeted correction [5]. Furthermore, there is a discussion on whether Achilles tendon lengthening is necessary as an adjunct to surgical intervention [57]. Although this clinical concern has not been extensively addressed, it holds promise for investigation through gait analysis methodologies.

This article centres on the examination of Flexible Flat Foot (FFF) during childhood and adolescence. Of particular interest is investigating the transition into adulthood to determine whether untreated flexible feet become more rigid and contribute to increased problems. All the currently available studies on flatfoot give only a snapshot into the kinematics. Therefore, longitudinal studies employing 3D kinematic analysis to track the progression of flatfeet in children are imperative. Such studies will facilitate a deeper understanding of how flatfoot biomechanics evolve over time, thereby informing early intervention strategies. This approach will effectively bridge the divide between childhood analysis and the gait analysis of adult feet, as reviewed by Buldt et al. [58].

This article specifically focuses on idiopathic FFF, acknowledging that there are various other origins beyond idiopathic, including overcorrected clubfeet, Down syndrome, other syndromes, and cerebral palsy. While some studies have explored the distinctions between idiopathic and overcorrected clubfeet [35], further investigation is warranted to delineate the variations comprehensively. This is particularly crucial if therapeutic approaches successful in idiopathic cases can be applied to other diagnoses.

The technology utilized in these studies was primarily based on marker-based models. Recently, supervised learning, a method within artificial intelligence (AI), has shown promise in markerless tracking of leg movement in healthy controls [59,60]. However, it is important to note that this method has been primarily trained on data from healthy individuals, potentially limiting its ability to accurately detect pathologic movements compared to marker-based systems [59]. To the best of our knowledge, markerless tracking of foot landmarks using AI has not yet been attempted. Previous attempts at markerless foot tracking involved scanning and fitting a visual hull to the foot model during walking. However, this approach was not successful in capturing the full gait cycle, particularly push-off and heel contact, which were challenging to accurately capture [61].

## 5. Conclusions

Instrumented 3D foot kinematic analysis is a valuable tool in the diagnosis and evaluation of the management of flatfeet in children and adolescents. The detailed insights into foot biomechanics during dynamic activities enhance diagnostic precision, guide treatment decisions, and contribute to a more holistic understanding of paediatric flatfeet. The incorporation of instrumented 3D kinematic analysis into routine clinical practice offers the potential for enhancing outcomes and elevating the quality of life for children affected by flatfeet.

## Figures and Tables

**Figure 1 children-11-00604-f001:**
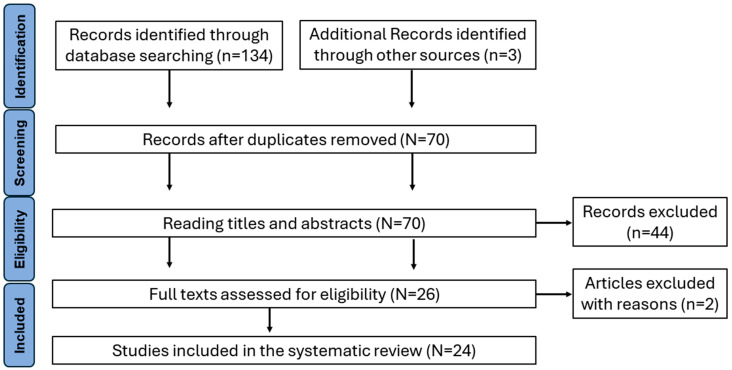
Article selection flowchart. Adapted from preferred reporting items for systematic review (PRISMA).

**Figure 2 children-11-00604-f002:**
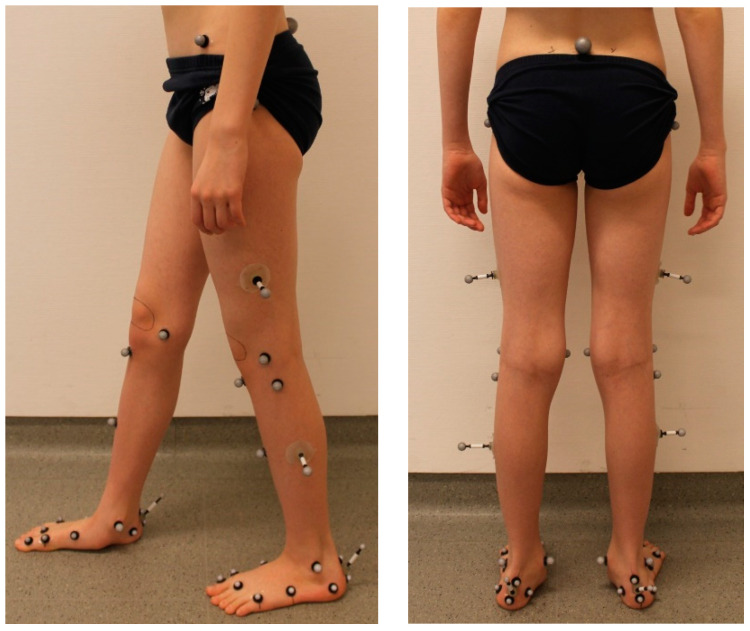
The patient with FFF was palpated using markers to assess the Plug-in Gait alongside the Oxford foot model.

**Figure 3 children-11-00604-f003:**
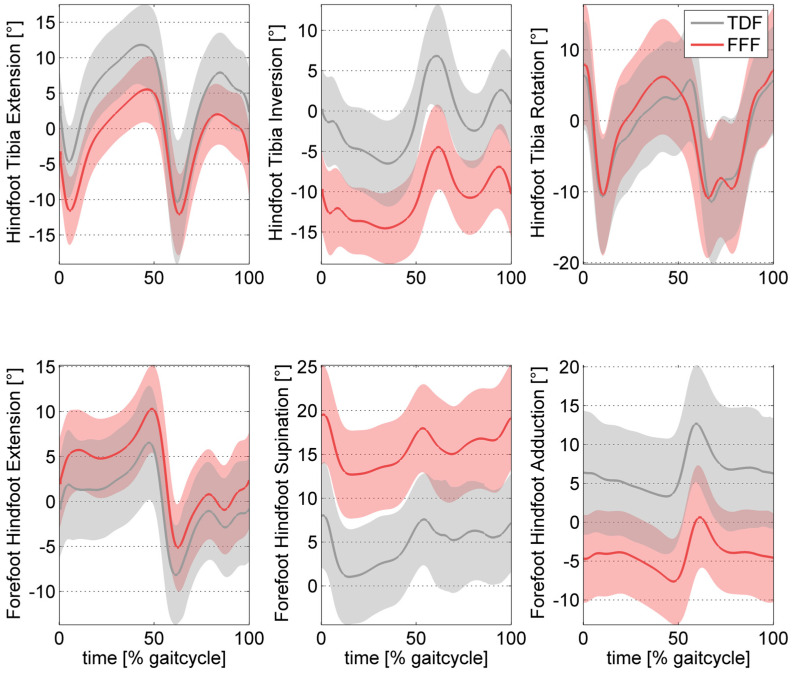
Typical deviations of severely affected flexible flatfeet (FFF) in three dimensions, compared to typically developed feet (TDF). The figure is based on the original data of Böhm et al. [31]. The FFF shown here were called decompensated by the authors since they did not show the muscular ability to perform an inversion of the hindfoot at push-off.

**Table 1 children-11-00604-t001:** Chronologicalal compilation of studies on 3D foot and lower body kinematics and kinetics in paediatric idiopathic FFF. Includes population description, measurement model, condition, study objectives, and key findings. Abbreviations: OFM (Oxford Foot Model), PiG (Vicon Plug-in Gait Model), BFSSS (Barefoot Walking at Self-selected Speed), TDF (Typically Developed Feet), and GRF (Ground Reaction Force).

Author, Year of Publication	Population of Idiopathic FFF	Models and Movement Conditions	Purpose	Main Results
Twomey et al., 2010 [17]	N = 27, age = 11.2 ± 1.2 years, asymptomatic, navicular height during stance phase < 20 mm. Selected from a sample of 94 feet recruited for another study without further information.	Heidelberg, BFSSS.	Comparative analysis between FFF and TDF.	Foot: Increased forefoot supination throughout the whole gait cycle. Lower body: N/A Kinetics: N/A
Twomey et al., 2012 [18]	N = 12, age = 12.2 ± 0.4 [11,12] years, asymptomatic, navicular height during stance phase < 20 mm. Subsample from above study.	PiG, BFSSS	Comparative analysis between FFF and TDF.	Foot: N/A Lower body: Increased external hip rotation and greater knee valgus angle in the low arched foot compared to TDF controls. Kinetics: N/A
Shih et al., 2012 [19]	N = 20, age = 9.7 ± 0.9 years. The FFF was defined by the Feiss line. Elementary school children.	Electromagnetic tracking, participants wore the same indoor shoes, walking speed was not reported.	Comparative analysis between FFF and TDF.	Foot: No differences in calcaneal angles between FFF and TDF controls. Lower body: Increased hip rotation range from initial contact to peak internal rotation and increased peak knee internal rotation compared to TDF. Kinetics: N/A
Hösl et al., 2014 [20]	N = 21, age = 11.0 ± 2.6 years, asymptomatic. N = 14, Age = 11.6 ± 2.0 years. Symptomatic. Patients presented to the clinics because of their FFF.	OFM, Modified PiG [21], BFSSS.	Classification of foot types symptomatic and asymptomatic FFF and TDF.	Foot: No differences in foot kinematics between symptomatic and asymptomatic feet. FFF showed increased eversion, and reduced DF of the rearfoot to tibia and increased DF, supination and abduction of the forefoot to the rearfoot compared to TDF controls Lower body: No differences in foot progression. Kinetics: During loading response Asymptomatic FFF absorbed more energy than TDF. The generated ankle joint energy at push-off showed a significant reduction in symptomatic vs. asymptomatic feet.
Kothari et al., 2014 [22]	N = 25, age = [8–15] years. FFF N = 25 Age 11.1 [10.0–12.2] TDF N = 26 Age 12.4 [9.4–13.9] (Same selection method as my other papers) Did not go into symptoms.	OFM with an additional navicular marker added.	Comparative analysis between FFF and TDF regarding navicular motion in different activities.	Foot: Flatfeet demonstrated reduced navicular drift compared to neutral footed children. No difference was observed in navicular drop between groups. Lower body: N/A Kinetics: N/A
Portinario et al., 2014 [23]	N = 10, age = 13.1 ± 0.8 years. Patients presented to the clinics because of their FFF but did not require surgical interventions.	Rizzoli, BFSSS.	Examination of the relationship between foot characteristics and clinical observation of rearfoot valgus to optimize the Rizzoli model for use in FFF.	Foot: To improve the correspondence with clinical observation of a valgus orientation of the calcaneus, an additional marker of the attachment area of the Achilles tendon was added. However, in 25% of patients this did not match the clinical observation of rearfoot valgus. A clearer definition of the position of this marker should be defined. Lower body: N/A Kinetics: N/A
Kerr et al., 2015 [24]	N = 15, age = 11.5 ± 2.2 years, symptomatic. Selected from patients records in the gait lab. N = 29, age 10.7 ± 3.5 [5–18] years, asymptomatic. They were recruited as part of a larger study to describe typical gait in children without further information.	OFM, barefoot static weightbearing.	Classification of foot types. Asymptomatic and symptomatic FFF.	Foot: Hindfoot eversion and forefoot abduction were much greater in the symptomatic population. Lower body: N/A Kinetics: N/A
Kothari et al., 2015 [25]	N = 42, age = 11.9 ± 2.0 years. Children were recruited from the paediatric orthopaedic clinic, orthotic clinic and from the community.	OFM, BFSSS.	Examination of the relationship between foot characteristics and quality of life.	Foot: Increased hindfoot eversion and forefoot supination during gait in FFF compared to TDF. Both parameters are also strongly related to lower quality of life scores. Lower body: N/A Kinetics: N/A
Kothari et al., 2016 [26]	N = 48, age = [8–15] years. The majority were recruited from the orthopaedic clinic and were referred because of their FFF.	PiG, BFSSS.	Examination of the relationship between foot characteristics and pain, finding predictors of hip and knee pain.	No kinematic or kinetic parameters associated with a flat foot posture were related to increased proximal joint pain but relations between FFF and various leg and kinetic parameters were shown. Foot: N/A Lower body: A flatter foot posture was associated with increased peak external pelvis rotation in late stance and was also associated with increased knee valgus in midstance. Kinetics: The flat foot posture was significantly associated with a reduction in the second peak of the vertical GRF, which concomitantly reduced late stance hip extension and knee varus and rotation moments.
Pothrat et al., 2015 [27]	N = 9, age = 8.2 ± 3.4 years Recruited children had a rearfoot valgus > 4° and a medial arch flattening when standing.	OFM and PiG, BFSSS.	Examination of the relationship between foot characteristics of ankle dorsiflexion of a single segment PiG model to amulltisegment OFM.	Foot: PiG Model showed at heelstrike ankle dorsiflexion and varus, whereas OFM showing plantar flexion and valgus. Lower body: N/A Kinetics: N/A
Krautwurst et al., 2016 [28]	N = 16, age = 6.4 ± 2.3 years, painless. N = 10, age = 8.0 ± 2.5 years, painful. They were presented to the clinics because of their FFF.	Heidelberg, barefoot heel raise test.	Examination of the relationship between foot characteristics during the heel raise test to distinguish pain free from painful feet.	Foot: No significant differences were found between the painful and painless groups. Lower body: N/A Kinetics: N/A
Jafarnezhadgero et al., 2017 [29]	N = 14, age = 10.2 ± 1.4 [8–12] years. Navicular drop > 10 mm.	PiG, participants wore the same sport shoes, at self-selected walking speed.	Assessment of the effectiveness of orthotic interventions on moments of ankle, knee, and hip joints.	Foot: N/A Lower body: N/A Kinetics: Foot orthoses can decrease the ankle evertor moment, knee and hip abductor moments, and hip flexor moment in the dominant lower limb.
Kim et al., 2017 [30]	N = 26, age = 9.5 [7–13] years. Recruited from the outpatient clinic.	Helen Hayes, Orthotrack software 6.6 (Motion Analysis Corp., Santa Rosa, CA, USA), BFSSS.	Comparative analysis between FFF and TDF that were healthy university students with a mean age of 21.3 years.	Foot: N/A Lower body: The range of plantarflexion during push-off was significantly reduced in the FFF compared to TDF controls. At midstance, the knee was significantly more flexed in FFF compared to TDF controls. Kinetics: The mean GRF during the push-off phase was significantly lower for FFF compared to TDF. This concomitantly reduced the mean ankle moment and power
Caravaggi et al., 2018 [31]	N = 20, age = 13.3 ± 0.8 years. Presented to the clinics because of their FFF.	Rizzoli, BFSSS.	Comparative analysis between FFF and TDF in midfoot kinematics.	Foot: The midtarsal joint was more dorsiflexed, everted, and abducted In FFF than TDF controls. and showed reduced sagittal-plane RoM. The tarso-metarsal joint was more plantarflexed and adducted, and showed larger frontal-plane RoM. The medial longitudinal arch showed larger RoM and was lower throughout the stance phase of the gait cycle. Lower body: N/A Kinetics: N/A
Caravaggi et al., 2018 [14]	N = 13, age = 11.3 ± 1.6 years at surgery. Scheduled for surgery because of their FFF.	Rizzoli, BFSSS.	Assessment of the effectiveness of surgical interventions. Two different arthroereisis implants were compared.	Foot: Both implants appear effective in restoring physiological alignment of the rearfoot; however, the endo-orthotic implant appeared more effective in restoring a more correct frontal-plane mobility of foot joints. Lower body: Knee valgus in stance was not different to TDF Kinetics: The second peak of the GRF is reduced in FFF preoperatively, whereas the first peak was increased compared to TDF.
Jafarnezhadgero et al., 2018 [32]	N = 30, age = [8–12] years. Separated into N = 15, orthoses, 10.5 ± 1.4 years, and N = 15 controls 10.4 ± 1.5 years. Recruited from orthopaedic specialists in the local community. Navicular drop > 10 mm, rearfoot eversion > 4° and arch height index < 0.31.	PiG, participants wore the same sport shoes, at self-selected walking speed.	Assessment of the effectiveness of orthotic interventions. In a randomized controlled study, the effects of 4-month treatment with arch support foot orthoses were compared to a placebo condition.	Foot: N/A Lower body: Improvements after 4 months in walking kinematics in maximum ankle internal rotation angle, maximum knee external, and internal rotation angles. Kinetics: Significant lower vertical GRF at push off between FFF and TDF at baseline. First peak of GRF was significantly reduced following 4-month orthotic therapy.
Kerr et al., 2019 [33]	N = 19, age = 11.4 ± 2.2 years, symptomatic, recruited from the paediatric orthopaedic clinic. N = 17, age = 9.6 ± 3.2 years, asymptomatic, recruited from the community.	OFM, BFSSS.	Classification of foot types in asymptomatic vs symptomatic FFF.	Foot: The symptomatic group having significantly increased forefoot abduction throughout the stance phase compared to the asymptomatic group. Lower body: The symptomatic FFF group exhibited significant differences compared to the TDF group, showing increased knee flexion angle by 5° and elevated knee valgus angle by 3° at midstance. Kinetics: N/A
Böhm et al., 2019 [34]	N = 129, age = 11.7 ± 2.1 years Patients presented to the clinics because of their FFF.	OFM, BFSSS.	Mathematical classification of foot types using 3D-foot kinematics.	Foot: Two clusters of feet could be identified, interpreted as compensated and decompensated feet. Hindfoot to tibia inversion at push-off was the most important discriminator for compensated feet. Deviations of decompensated FFF compared to TDF could be observed in all 3 planes and rearfoot to tibia and forefoot to rearfoot with the largest deviations in the frontal plane rearfoot eversion and forefoot supination. Lower body: N/A Kinetics: N/A
Böhm et al., 2020 [35]	N = 204, age = 11.7 ± 1.9 years Patients presented to the clinics because of their FFF.	OFM, BFSSS.	Examination of the relationship between foot characteristics and radiography.	Foot: Three-dimensional foot kinematics showed little to no relation to radiographic measures. Lower body: N/A Kinetics: N/A
Kim et al., 2020 [36]	N = 22, 10.8 ± 1.51 years at surgery. Scheduled for surgery because of their FFF.	Helen Hayes, Orthotrack software (Motion Analysis Corp.), BFSSS.	Assessment of the effectiveness of surgical interventions. Calcaneal lengthening procedure was analysed before and 1 year following surgery.	Foot: Ankle valgus angle in the coronal plane was reduced from 35.5° preoperatively to 16.6° postoperatively. Lower body: The preoperative foot progression angle of 20° was normalized to 14° postoperatively. Kinetics: The push-off moment increased from 0.66 Nm/kg preoperatively to 0.83 Nm/kg postoperatively
Byrnes et al., 2021 [37]	N = 103, age = 11.7 ± 2.3 years. Patients presented to the clinics because of their FF. Of those N = 19, 11.3 ± 1.9 years underwent surgeries and follow up.	OFM, Modified PiG [38], BFSSS.	Examination of the relationship between foot characteristics with the knee adduction moments (KAM).	Foot: N/A Lower body: Knee valgus angle was not significantly different between FFF and TDF controls. It should be mentioned that children with knee varus/valgus deformities and in- and out-toeing were excluded from the study. Kinetics: Lateral calcaneal shift and arch height correlated with KAM. Only the change in lateral calcaneal shift correlated to the change in KAM following surgery. Children with FFF hat significantly lower peak KAM in the first and second half of stance compared to TDF.
Boryczka-Trefler et al., 2022 [39]	N = 49, Age= 6.4 [5.0–10.4] years Presented to the outpatient clinic with an established clinical FFF.	PiG, BFSSS.	Comparative analysis between FFF and TDF: Does the method (static vs. dynamic) of assessing FFF severity influence lower limb kinematic differences compared to TDF?	Only the discrimination by dynamic arch index of >0.27 leads to the following significant differences during walking: Foot: N/A Lower body: The pelvic rotation and ankle ROM was smaller in more severe flatfeet. Kinetics: The maximal values of vertical GRF components in the middle of stance were larger and during push-off were smaller in FFF than in TDF.
Böhm and Dussa 2023 [5]	N = 177, age = 11.8 ± 2.2 [7–17] years. Presented to the clinics because of their FFF. Of those N = 31, 11.2 ± 1.4 years underwent arthroereisis surgeries and follow up.	OFM, BFSSS.	Examination of the relationship between foot characteristics and medial arch pain and the relation to the reduction in pain following surgical treatment.	Foot: Pain was perceived in 52% of the feet, of these, 74% was in the medial arch. The calcaneal lateral shift during walking showed a significant difference between the no pain and pain groups and was associated with the reduction in pain following surgery. Lower body: N/A Kinetics: N/A
Pourghazi et al., 2023 [40]	N = 7, age = 12.2 ± 2.9 years, Scheduled for surgery because of their FFF.	OFM, BFSSS.	Assessment of the effectiveness of surgical intervention calcaneal lengthening osteotomy before and 6 months after surgery.	Foot: Not reported because of volatile results, e.g., ankle varus in FFF and valgus in TDF. Lower body: External foot progression angle was improved from 11.3 ± 6 to 16.2 ± 7.1 after surgery. The (1 maximum plantar flexion decreased. Kinetics: Peak ankle moments and powers of FFF patients are significantly smaller than TDF. Following surgery ankle moment and power were not significantly different.

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
