# Peer review of "Dynamic Gait Analysis in Paediatric Flatfeet: Unveiling Biomechanical Insights for Diagnosis and Treatment"

_children, 2024, doi:10.3390/children11050604_

Round 1
Reviewer 1 Report
Comments and Suggestions for Authors
This systematic review without meta-analysis is very well conducted and presented. The methods for study selection and inclusion are clear and appropriate. The analysis approach reflects the diverse purposes of the various included studies. The writing is clear, organized, and thoughtful. The authors should consider re-characterizing this paper as a systematic review without meta-analysis rather than as a narrative review. The former better reflects the rigor and standardization used to identify studies for inclusion in this review.
Line 65. Provide a brief explanation for why studies were restricted to 2010 and later.
Lines 82. Provide a description of how results were assessed. Was this done by consensus discussion among the authors? By individual authors taking the lead in certain sections? What procedures were adopted to assure that the summaries provided adequately reflect the published results?
Table 1. This table is extremely helpful, though the Purpose column could be enhanced. Clearly state for each study whether it investigated FFF only or compared FFF to TDF. Also consider adding to this column how the review authors categorized that study’s purpose using the key areas of investigation outlined in lines 128-132. In the headers for the table, relabel the “Models and conditions” table to be “Models and Movement Conditions” so the reader recognizes there may be differences across the studies in what movement was performed or under what conditions.
The Results, Discussion and Conclusion were well described and comprehensive.
Author Response
Authors: We extend our heartfelt thanks for taking the time to review our article. Your insights were invaluable in refining our work, and we deeply appreciate your constructive feedback.
Reviewer 1: This systematic review without meta-analysis is very well conducted and presented. The methods for study selection and inclusion are clear and appropriate. The analysis approach reflects the diverse purposes of the various included studies. The writing is clear, organized, and thoughtful. The authors should consider re-characterizing this paper as a systematic review without meta-analysis rather than as a narrative review. The former better reflects the rigor and standardization used to identify studies for inclusion in this review.
Authors answer: Thank you for your kind words and suggestion to elevate the review to a systematic level. Your input is greatly appreciated.
Authors action: We adhered to the PRISMA guidelines and included the necessary descriptions and a flowchart in our submission. Regrettably, due to time constraints (5 days), we couldn't conduct an assessment of the paper quality utilizing methods such as the Downs and Black Epidemiol Community Health 1998;52:377–84 scoring system. Given that the focus of the review does not center on a treatment method, applying this assessment might pose challenges. However, should it be deemed necessary, we are prepared to incorporate this procedure upon request.
Reviewer 1: Line 65. Provide a brief explanation for why studies were restricted to 2010 and later.
Authors answer: The popularity of multisegment foot models surged around 2010, influenced by Theologis and Stebbins' seminal paper on the use of gait analysis in pediatric foot and ankle disorders (Foot Ankle Clin, 2010 Jun;15(2):365-82). Notably, prior to 2010, there is a dearth of literature utilizing multisegment foot models in flatfeet research.
Reviewer 1: Lines 82. Provide a description of how results were assessed. Was this done by consensus discussion among the authors? By individual authors taking the lead in certain sections? What procedures were adopted to assure that the summaries provided adequately reflect the published results?
Authors action: the study selection process was added in greater detail in section 2.3. and the quality assurance procedure to synthesize the summaries in section 2.4.
Reviewer 1: Table 1. This table is extremely helpful, though the Purpose column could be enhanced. Clearly state for each study whether it investigated FFF only or compared FFF to TDF. Also consider adding to this column how the review authors categorized that study’s purpose using the key areas of investigation outlined in lines 128-132. In the headers for the table, relabel the “Models and conditions” table to be “Models and Movement Conditions” so the reader recognizes there may be differences across the studies in what movement was performed or under what conditions.
Authors answer: it is an excellent idea to use the 4 classifications in describing the purpose of the studies.
Authors action: One of the four classification was added to each of the studies. To the second header “movement” was added as suggested.
Reviewer 1: The Results, Discussion and Conclusion were well described and comprehensive.
Authors answer: thank you very much.
Reviewer 2 Report
Comments and Suggestions for Authors
Dear Author,
Thank you for the opportunity to review this article.
Each abbreviation should be explained in brackets in the first mention (I.E. GRF). Once abbreviated, there is no need to explain the word again (like in line 86). TDF should be explained at the first contact, not in row 137.
In the table, you use parentheses and brackets interchangeably. Choose one and keep it the same.
The introduction is adequate.
Please explain TW in the search strategy.
Minor Spelling errors need to be addressed (I.E. line 213)
In Challenges and future directions, we suggest adding a phrase about subtalar arthroereisis and, if possible, the kinematic indications. Here is a suggested article: 10.3390/children9070973.
Please clearly rewrite the conclusions. First 2 phrases are adequate, but the third one is more appropriate to a discussion.
Author Response
Authors: We wish to express our sincere appreciation for taking the time to review our article. Your insightful feedback and helpful suggestions have been valuable to us.
Reviewer 2: Each abbreviation should be explained in brackets in the first mention (I.E. GRF). Once abbreviated, there is no need to explain the word again (like in line 86). TDF should be explained at the first contact, not in row 137.
Authors answer: we apologize for not introducing the abbreviations appropriately.
Authors action: following the APA guidelines we added all abbreviations that were used in the table 1 to the caption ot the table 1. It was not clearly written in the guideline; however, our interpretation was that abbreviations should also be defined in the main text upon its first appearance, although they were defined in a table caption before. That was e.g. in case of TDF in line 130. In addition, Abbreviations were avoided in the headings and description of figures. This process was reviewed again for the abbreviations FFF, TDF, PiG, OFM, GRF.
Reviewer 1: In the table, you use parentheses and brackets interchangeably. Choose one and keep it the same.
Authors action: We reviewed the table 1 and changed the age range into brackets consistently, other parentheses were removed.
Reviewer 1: Please explain TW in the search strategy.
Authors action: an explanation of [tw] = textword was added to section 2.2 search strategy.
Reviewer 1: Minor Spelling errors need to be addressed (I.E. line 213)
Authors action: error in line 213 was corrected.
Reviewer 1: In Challenges and future directions, we suggest adding a phrase about subtalar arthroereisis and, if possible, the kinematic indications. Here is a suggested article: 10.3390/children9070973 .
Authors action: the potential application to indication of arthroereisis surgery was added to section 4.5 Challenges and future directions.
Reviewer 1: Please clearly rewrite the conclusions. First 2 phrases are adequate, but the third one is more appropriate to a discussion.
Authors action: we removed the aspect of technology advancement from the conclusion, since this was not investigated in the review.